# Thermal remote sensing over heterogeneous urban and suburban landscapes using sensor-driven super-resolution

**Hiroki Mizuochi**[ID]*, **Koki Iwao**[☯], **Satoru Yamamoto**[☯]

Geological Survey of Japan, National Institute of Advanced Industrial Science and Technology, Tsukuba, Ibaraki, Japan

☯ These authors contributed equally to this work.

* mizuochi.hiroki@aist.go.jp

**Data Availability Statement:** The data analyzed in the paper are freely available on the JAXA website (https://www.eorc.jaxa.jp/ALOS/a/en/dataset/lulc/lulc_jpn_e.htm), DAAC website (MODIS, https://ladsweb.modaps.eosdis.nasa.gov/search/), and

## Abstract

Thermal remote sensing is an important tool for monitoring regional climate and environment, including urban heat islands. However, it suffers from a relatively lower spatial resolution compared to optical remote sensing. To improve the spatial resolution, various "data-driven" image processing techniques (pan-sharpening, kernel-driven methods, and machine learning) have been developed in the previous decades. Such empirical super-resolution methods create visually appealing thermal images; however, they may sacrifice radiometric consistency because they are not necessarily sensitive to specific sensor features. In this paper, we evaluated a "sensor-driven" super-resolution approach that explicitly considers the sensor blurring process, to ensure radiometric consistency with the original thermal image during high-resolution thermal image retrieval. The sensor-driven algorithm was applied to a cloud-free Moderate Resolution Imaging Spectroradiometer (MODIS) scene of heterogeneous urban and suburban landscape that included built-up areas, low mountains with a forest, a lake, croplands, and river channels. Validation against the reference high-resolution thermal image obtained by the Advanced Spaceborne Thermal Emission and Reflection Radiometer (ASTER) shows that the sensor-driven algorithm can downscale the MODIS image to 250-m resolution, while maintaining a high statistical consistency with the original MODIS and ASTER images. Part of our algorithm, such as radiometric offset correction based on the Mahalanobis distance, may be integrated with other existing approaches in the future.

## Introduction

Measurement of terrestrial thermal emissions allows us to estimate the land surface temperature and the emissivity of surface materials. Thermal remote sensing takes advantage of such features to effectively monitor volcanic disasters [1], wildfires [2], crop fields [3], mineral composition [4], regional climate [5] and urban heat islands [6, 7]. In comparison to observation using in situ photographs [8] or unmanned aerial vehicles [9], satellite-based observation has advantages in spatial coverage, frequency, and regularity.

AIST MADAS websites (ASTER, https://gbank.gsj.jp/madas/?lang=en#top).

**Funding:** The authors received no specific funding for this work.

**Competing interests:** The authors have declared that no competing interests exist.

One of the major issues in thermal remote sensing is the coarse spatial resolution of the thermal images [4]. In comparison to optical sensors that observe solar reflection from the Earth's surface, thermal sensors that observe thermal emissions from the surface collect electromagnetic waves with lower signal strength, resulting in lower spatial resolution. For example, the spatial resolution of the optical data provided by the Moderate Resolution Imaging Spectroradiometer (MODIS) is 250 m or 500 m, whereas that of thermal data is 1 km. A similar situation arises for other moderate resolution sensors: the resolution of optical data provided by the Advanced Spaceborne Thermal Emission and Reflection Radiometer (ASTER) is 15 m or 30 m, but the thermal data resolution is 90 m.

The other aspect of degraded spatial resolution is the general trade-off between spatial and temporal resolution, and the spatial and spectral resolution in a single sensor. Due to technical limitations (especially in data downlinks), frequent and/or spectrally fine-resolution observation sacrifices spatial details, and vice versa. Missing spatial detail in thermal images is particularly critical when monitoring heterogeneous landscapes, such as urban and suburban areas.

To enhance the spatial resolution of thermal images, a wide variety of techniques, called "disaggregation," "downscaling," and "super-resolution," have been developed in recent decades [10]. These can be roughly divided into multi-sensor-based and single-sensor-based approaches. The multi-sensor-based approach, also called spatiotemporal image fusion [11], mainly focuses on mitigating the trade-off between spatial and temporal resolution. In this approach, thermal images with high spatial (but low temporal) resolution are estimated from simultaneously (or quasi-simultaneously) acquired thermal images with low spatial (but high temporal) resolution, based on an empirical relationship between them. Various algorithms, such as the spatial and temporal adaptive reflectance fusion model [12] and similar or improved models (e.g., [13–16]), are used to describe the relationship. These algorithms are powerful tools for environmental monitoring with high spatiotemporal resolution, and are widely applied with match-up pairs between MODIS and ASTER [17], MODIS and Landsat [15], and polar orbiting satellites and geostationary satellites [14]. However, given the nature of spatiotemporal image fusion, the success or failure of this approach depends on the selection of the matched pairs used to describe the relationship.

In contrast, the single-sensor-based approach relies on a relationship between the thermal image and images in other spectral domains (usually optical) acquired by the same instrument, to enhance the spatial resolution of thermal images. This approach can be applied to a single sensor that observes thermal and another spectral domain simultaneously from the same platform, even in the absence of a counterpart satellite platform that offers a sufficient chance of simultaneous overpasses of the region of interest, which is rarely realized for satellites with irregular orbits, and for deep space exploration. Pan-sharpening methods via intensity-hue-saturation transformation or principal component analysis have been used traditionally [18, 19], and kernel-driven methods [20–22] and machine-learning approaches (e.g., [23–25]) have become popular recently. These efforts have created visually appealing thermal images that have higher spatial resolution than the original ones. However, such "data-driven" approaches do not necessarily take physical processes into account, including sensor-specific features, and radiometric consistency.

In contrast, there are a few "sensor-driven" approaches that explicitly consider sensor features, and target radiometric consistency in the super-resolution results. Hughes and Ramsey [4] introduced a sensor-driven super-resolution approach originally developed by Tonooka [26], which creates both quantitatively accurate and qualitatively acceptable results for their exploration of Mars using the Thermal Emission Imaging System (THEMIS) onboard the Mars Odyssey [27]. This simple approach uses the Mahalanobis distance to estimate each high-resolution pixel value from neighboring, spectrally similar low-resolution pixels.

Beneficial characteristics of this approach include consideration of the point spread function (PSF) for the sensor of interest, and radiometric correction weighted by the Mahalanobis distance after the tentative super-resolution retrieval. Such an approach that gives attention to sensor physics also seems to be in line with the recent trend of physically informed machine learning [28], and worth revisiting to achieve single-sensor-based super-resolution rather than using the empirical, data-driven approaches [18–25]. However, sufficient validation and evaluation of the super-resolution results obtained using the sensor-driven algorithm over a heterogeneous Earth surface have not been conducted. In addition, as the original algorithm was proposed more than 10 years ago [26], there seems to be room for refinement. Although it was implemented with ASTER data over urban and suburban areas, quantitative accuracy assessments using independent validation data have not been provided yet.

Therefore, this work aims to investigate the potential applicability of the sensor-driven approach over a heterogeneous landscape, and to improve its primitive algorithm. A complex terrain including urban, suburban, forest, lake, and river areas was selected as the study site for this purpose. Similar to previous thermal super-resolution research (e.g., [17, 22]), Terra/MODIS was used as the sensor of interest. The relatively wide swath of MODIS is suited to covering large areas and capturing various land cover types in comparison with other moderate-resolution instruments (e.g., Landsat) that are also often used for super-resolution algorithm development. The other advantage of Terra/MODIS is the existence of a counterpart higher-resolution sensor (ASTER), which can be used for validation data. Because they are onboard the same satellite platform and make simultaneous observations, comparison between them can minimize differences in atmospheric and/or surface conditions [29]. Because both MODIS and ASTER data are freely available, readers can easily reproduce our results. The radiometric calibration uncertainty (sensor requirement) for MODIS thermal bands for surface temperature measurement (i.e., bands 31 and 32) is ± 0.5% in radiance [30]. That for ASTER is ± 1 K or better in brightness temperature, for the range of 270–340 K (i.e., ~ ± 0.3%) [31]. In-flight validation of the thermal bands of MODIS and ASTER has also been reported by Hook et al. [32]. This work provides the first quantitative accuracy assessment of sensor-driven super-resolution with MODIS, using independent validation data (ASTER).

## Materials and methods

We first describe the original algorithm developed by Tonooka in 2005 [26] in the "Original algorithm" section, and then describe our proposed refinement in "Proposed refinement" section. Descriptions of the study site and data are provided in the "Study site and data processing" section.

### Original algorithm

The original algorithm for the sensor-driven approach was proposed by Tonooka [26]. It is a single-sensor-based approach, and thus makes full use of high-resolution optical information to achieve super-resolution with the low-resolution thermal pixels. It relies on "the empirical fact that, if two nearby surfaces are covered by a similar material under a similar situation, their radiance spectra will be similar in the wide wavelength region" [26]. Therefore, application of the algorithm is not limited to correlation of thermal and optical images. As long as the abovementioned assumption is reasonable, the algorithm is applicable (and was actually applied), even between visible and near infrared bands and shortwave infrared bands.

For a general description, let us denote a pixel value of a higher-resolution image in band $k$ (= 1, 2, . . ., $n$) as $f_{high,k}$, and that of the counterpart lower-resolution image in band $k'$ ($k'$ = 1, 2, . . ., $m$) as $g_{low,k'}$. By an appropriate inter-band coregistration and reasonable sensor design, we

assume that one lower-resolution pixel corresponds to an integer number of higher-resolution pixels. The overall super-resolution procedure is as follows:

Step 1) Search homogeneous pixels within each lower-resolution scale.

Step 2) Degrade $f_{\text{high,k}}$ images to the same resolution of $g_{\text{low,k'}}$ images considering the PSF (denoted as $f_{\text{low,k}}$ hereafter).

Step 3) Make a typical spectral pattern (i.e., correspondence between $f_{\text{low,k}}$ and $g_{\text{low,k'}}$) by clustering the homogeneous pixels within the entire study region.

Step 4) Calculate the Mahalanobis distance ($d_{\text{nei}}$) between $f_{\text{high,k}}$ at the pixel of interest and $f_{\text{low,k}}$ at neighboring homogeneous pixels within a moving window.

Step 5) Calculate the Mahalanobis distance ($d_{\text{lib}}$) between $f_{\text{high,k}}$ at the pixel of interest and the typical spectral pattern extracted in step 3.

Step 6) Compare all Mahalanobis distances calculated in steps 4 and 5, and assign $g_{\text{low,k'}}$ at the minimum $d_{\text{nei}}$ or $d_{\text{lib}}$ as the super-resolved pixel value ($g_{\text{high,k'}}$).

Step 7) Repeat steps 4–6 for all high-resolution pixels.

Step 8) Add an offset so that degraded $g_{\text{high,k'}}$ can be consistent with the original $g_{\text{low,k'}}$ for each low-resolution pixel (i.e., perform radiometric correction). The offset value is determined for each high-resolution pixel from the Mahalanobis distance and PSF.

Fig 1 summarizes the super-resolution steps in the form of a flowchart. The image pairs for (A) high-resolution bands and (B) low-resolution bands are input into the process. The high-

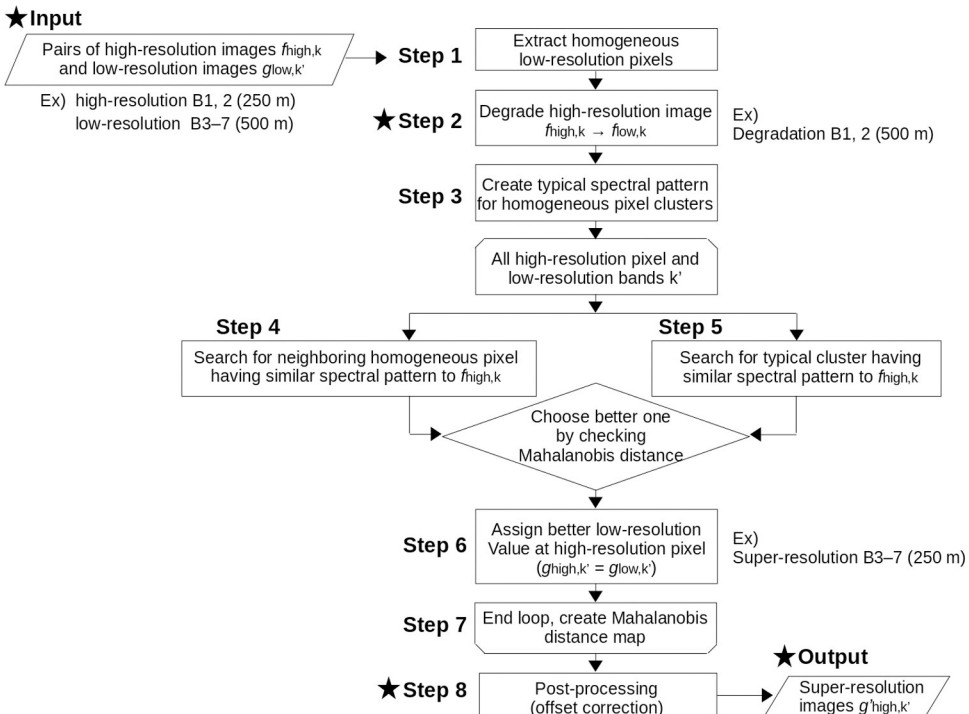

**Fig 1. Flowchart for super-resolution process.** The star symbols are where our refinement from the original algorithm [26] was implemented. As an example, the super-resolution process for converting MODIS 500-m resolution images (band 3–7) to 250-m resolution images is shown.

resolution images are degraded to the same resolution as the low-resolution images in step 2. For each high-resolution pixel location, the B data is positioned by referring to the relationship between the A and B spectral information, either at a neighboring homogeneous pixel (step 4) or in a typical spectral pattern (step 5). By repeating this process (step 6) for all high-resolution pixel locations (step 7), images having B-band information but having A-band spatial resolution are created (i.e., super-resolution). The final result is output after post-processing (step 8). A more detailed explanation of each step is provided in the following section.

Theoretically, this process can be applied to any two datasets that have different spatial resolutions, as long as they have some statistical relationship. In the case of MODIS, there are terrestrial bands with three different spatial resolutions (i.e., 250 m for bands 1 and 2, 500 m for bands 3–7, and 1 km for thermal bands), leading to arbitrariness in combining these bands to obtain super-resolution. In the case of the original algorithm [26], band 3–7 (500-m resolution) were first super-resolved to a resolution of 250 m by referring to the highest-resolution bands (bands 1 and 2), and then the thermal bands (1-km resolution) were super-resolved to a resolution of 250 m by referring to bands 1 and 2, and previously obtained super-resolution bands (3–7).

The input-output process for this "two-times super-resolution" method is shown in Fig 2. In the flowchart (Fig 1), original bands 1 and 2 (250-m resolution) correspond to $f_{high,k}$, degraded bands 1 and 2 (500-m resolution) correspond to $f_{low,k}$, which are indicated by the two red arrows input to the first super-resolution step in Fig 2. The original bands 3–7 (500-m resolution) correspond to $g_{low,k'}$, which is shown as the blue arrow input to the first super-resolution step. The super-resolved bands 3–7 (250-m resolution) are further input to the second super-resolution step with the original bands 1–2 ($f_{high,k}$ in the second super-resolution step), as well as both degraded bands (1-km resolution; $f_{low,k}$) and the original thermal bands ($g_{low,k'}$). The final output is the thermal images (bands 31, 32) with a resolution of 250 m.

Note that band 5 of Terra/MODIS suffers from stripe noise [33], and we decided not to use it for further processing.

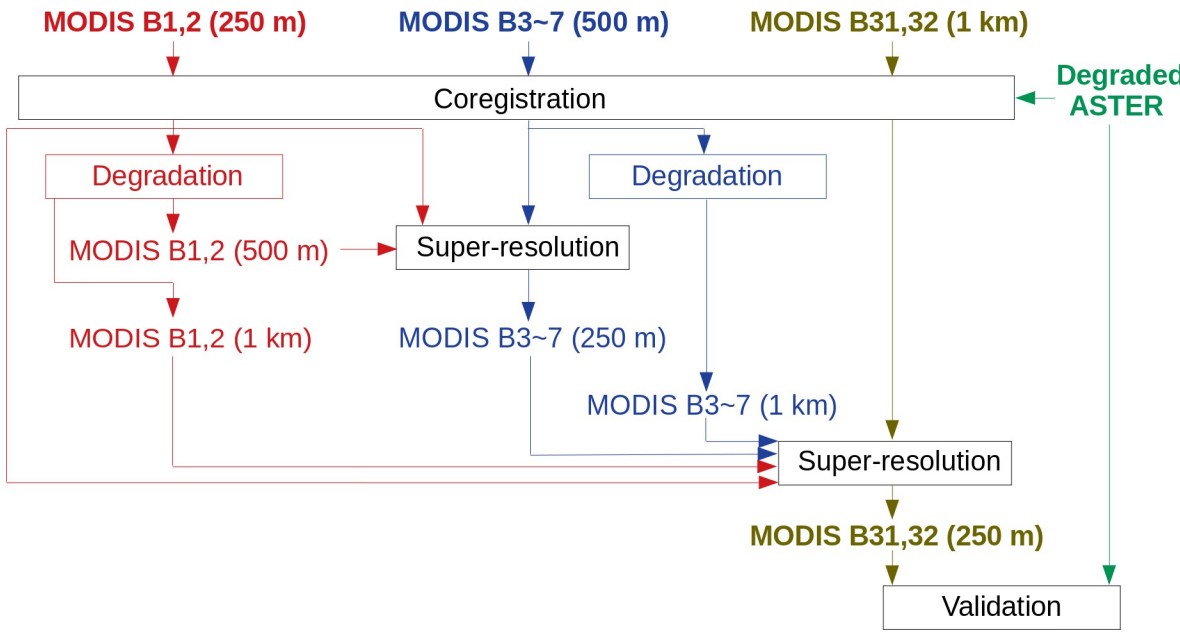

**Fig 2. Original super-resolution algorithm proposed by Tonooka in 2005 [26].**

For the second super-resolution process, the Mahalanobis distance from the highest-resolution bands and the previously super-resolved bands (bands 3–7 in our case) were calculated separately. The total Mahalanobis distance is evaluated by

$$d_{\text{total}}^2 = \frac{w}{n_1} d_1^2 + \frac{1-w}{n_2} d_2^2, \tag{1}$$

where $d_1$ is the Mahalanobis distance (either $d_{\text{nei}}$ or $d_{\text{lib}}$) for bands 1 and 2 in our case, $d_2$ is that for bands 3–7, $n_1$ (= 2) and $n_2$ (= 4) are the corresponding number of bands, and $w$ is a weighting factor, which was assumed to be 0.7 [26].

## Proposed refinement

To make the algorithm more straightforward and to create better radiometrically corrected results, in this paper, we propose two modifications regarding (1) the order of multiple super-resolution retrievals and (2) regularization of the offset adjustment. For each super-resolution process, refinement (1) concerns input-output correspondence and degraded image input, whereas refinement (2) concerns post-processing (both are indicated by a star symbol in the flowchart in Fig 1).

For the first modification, the second-highest resolution images are first super-resolved to the highest resolution, which are used in the second super-resolution process in the original algorithm. In this case, the first super-resolution process relies only on the two highest-resolution bands (bands 1 and 2), which is likely to cause substantial uncertainty in the first super-resolution retrieval. The uncertainty probably propagates to the second super-resolution retrieval, making it difficult to perform reliable analysis with the super-resolution results. In addition, regarding this procedure, the original algorithm evaluates the Mahalanobis distance from the highest-resolution bands and super-resolves the second-highest resolution bands separately (Eq 1). This seems to make the algorithm complex and requires the somewhat arbitrary hyperparameter $w$.

To avoid this complexity, we applied the procedure in the inverse direction: first, thermal bands were super-resolved to 500 m with the help of bands 1–7, the result of which was further super-resolved to 250 m with the help of bands 1 and 2 (Fig 3). The MODIS bands 1 and 2 were degraded to 500 m and 1 km, and bands 3–7 were degraded to 1 km in the first super-resolution retrieval. In other words, bands 1–7 (500-m resolution) were $f_{\text{high,k}}$, bands 1–7 (1-km resolution) were $f_{\text{low,k}}$, and bands 31, 32 were $g_{\text{low,k'}}$, which were all input to the first super-resolution step. These were used together for calculation of the Mahalanobis distance, and thus Eq 1 and the arbitrary parameter $w$ were no longer needed. The procedure enables the first super-resolution retrieval to make full use of all the optical bands, which may also improve the second super-resolution retrieval and yield a more reliable final result.

For this modification, a detailed description of each step of the algorithm is provided below.

Step 1) Within each low-resolution pixel, the standard deviation of $f_{\text{high,k}}$ is calculated. Homogeneous pixels are flagged when the standard deviation within a low-resolution pixel exceeds the standard deviation over the entire study area for all bands $k$. In the first super-resolution process, $k$ ranges from band 1 to 7 with 500-m resolution (i.e., $n = 7$), whereas in the second super-resolution process, $k$ ranges from band 1 to 2 with 250-m resolution (i.e., $n = 2$).

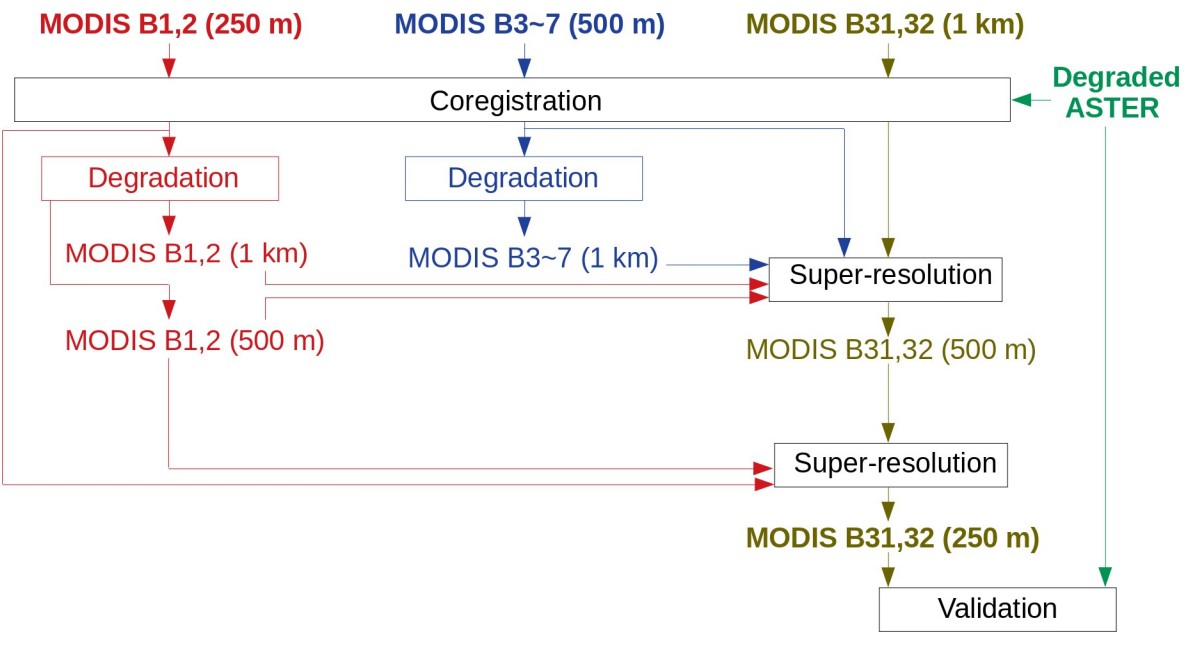

**Fig 3. Proposed super-resolution algorithm.**

Step 2) Within each low-resolution pixel, $f_{high,k}$ is degraded using the PSF as a weighting function to describe signal blurring in low-resolution sensor observation:

$$f_{low,k} = \sum\nolimits_{(i,j) \subseteq lowres} PSF(i,j) \times f_{high,k}(i,j), \qquad (2)$$

where $i$ and $j$ denote high-resolution pixel locations within a low-resolution pixel. The mathematical expression for the PSF is provided in the "Study site and data processing" section.

Step 3) For all homogeneous pixels, K-means++ clustering is conducted first with $f_{low,k}$, and then with $g_{low,k'}$. The number of clusters is set to nine based on the land cover types of the study site (see "study site and data processing" section). The clusters for $f_{low,k}$ and $g_{low,k'}$ compose hierarchical trees. For each node of the tree, samples of $f_{low,k}$ and $g_{low,k'}$ are averaged and stored as a database, creating typical spectral patterns over the entire study region.

Step 4) Homogeneous pixels are picked up within ±10 low-resolution pixels (i.e., a moving window) from the high-resolution pixel of interest. The Mahalanobis distance is calculated by

$$d_{nei}^2 = (\mathbf{f}_{high} - \mathbf{f}_{low})^T \mathbf{V}_f^{-1} (\mathbf{f}_{high} - \mathbf{f}_{low}), \qquad (3)$$

where $\mathbf{f}_{high} = (f_{high,1}, f_{high,2}, \ldots, f_{high,n})^T$ is a vector with pixel values at the pixel of interest, $\mathbf{f}_{low} = (f_{low,1}, f_{low,2}, \ldots, f_{low,n})^T$ is a vector of homogeneous pixels, and $\mathbf{V}_f$ is a variance-covariance matrix of $\mathbf{f}_{low}$ for all the homogeneous pixels of the study site. The homogeneous pixel with minimum $d_{nei}$ (i.e., the spectrum most similar to the pixel of interest) is a candidate for $g_{high,k'}$.

Step 5) Similarly, the Mahalanobis distance for the typical spectral pattern is calculated by

$$d_{\text{lib}}^2 = (\mathbf{f}_{\text{high}} - \mathbf{f}_{\text{lib}})^T \mathbf{V}_f^{-1} (\mathbf{f}_{\text{high}} - \mathbf{f}_{\text{lib}}), \tag{4}$$

where $\mathbf{f}_{\text{lib}}$ is a column vector with the average pixel values (band $k = 1, 2, \ldots, n$) from each cluster. The minimum $d_{\text{lib}}$ is also a candidate to estimate $g_{\text{high},k'}$.

Step 6) The minimum $d_{\text{nei}}$ and the minimum $d_{\text{lib}}$ are compared. When $d_{\text{nei}} \leq d_{\text{lib}}$, $g_{\text{low},k'}$ at the homogeneous pixel with the minimum $d_{\text{nei}}$ is placed into the high-resolution pixel as $g_{\text{high},k'}$. When $d_{\text{nei}} > d_{\text{lib}}$, the algorithm searches for the spectrum in the $g$ domain at the node where $d_{\text{lib}}$ was a minimum:

$$d_{\text{lib},g}^2 = (\mathbf{g}_{\text{low}} - \mathbf{g}_{\text{lib}})^T \mathbf{V}_g^{-1} (\mathbf{g}_{\text{low}} - \mathbf{g}_{\text{lib}}), \tag{5}$$

where $\mathbf{g}_{\text{low}} = (g_{\text{low},1}, g_{\text{low},2}, \ldots, g_{\text{low},m})^T$ is a vector with pixel values at the pixel of interest, $\mathbf{g}_{\text{lib}}$ is a column vector with average pixel values from each $g$ cluster at the node with minimum $d_{\text{lib}}$, and $\mathbf{V}_g$ is a variance-covariance matrix of $\mathbf{g}_{\text{low}}$ for all the homogeneous pixels in the study site. The average $g_{\text{low},k'}$ at the node of minimum $d_{\text{lib},g}$ is placed as $g_{\text{high},k'}$.

Step 7) Steps 4–6 are repeated for all high-resolution pixels to create a high-resolution $g$ image with $g_{\text{high},k'}$. At the same time, the Mahalanobis distance corresponding to the adopted $g_{\text{high},k'}$ is stored for each pixel as a "distance map."

Step 8) The retrieved $g_{\text{high},k'}$ should be radiometrically consistent with $g_{\text{low},k'}$ when degraded again within a low-resolution pixel. To this end, an offset value is added to $g_{\text{high},k'}$. Instead of adding an offset uniformly over the low-resolution pixels, full use is made of the Mahalanobis distance, to allow additional offset corrections to be made for less reliable pixels of $g_{\text{high},k'}$ (i.e., pixels with less spectral similarity). The offset to meet this concept is

$$g_{\text{high},k'}' = g_{\text{high},k'} + \alpha_{k'} \times d^2, \tag{6}$$

where $d$ is the Mahalanobis distance from the distance map, $g'_{\text{high},k'}$ is the corrected result, and $\alpha_{k'}$ is a modification scale defined by

$$\alpha_{k'} = \frac{g_{low,k'} - \sum_{(i,j) \subseteq \text{lowres}} PSF(i,j) \times g_{high,k'}(i,j)}{\sum_{(i,j) \subseteq \text{lowres}} PSF(i,j) \times d^2(i,j)}. \tag{7}$$

Our second modification of the original algorithm regards the offset correction. The above-mentioned offset correction with consideration of the Mahalanobis distance as a weighting function is theoretically reasonable; however, a very large Mahalanobis distance among a few pixels may result in overcorrection and implausible pixel values. To mitigate overcorrection while still employing the concept of weighting by the Mahalanobis distance, we introduced a regularization term into the distance map:

$$d_{\text{norm}}^2 = \frac{d^2}{\sum_{(i,j) \subseteq \text{ALL}} d^2(i,j)}, \tag{8}$$

$$d_{\text{reg}}^2 = d_{\text{norm}}^2 + \lambda, \tag{9}$$

where $d_{\mathrm{norm}}$ is a normalized distance that makes the summation over the entire study region equal to 1, $d_{\mathrm{reg}}$ is the regularized distance, and $\lambda$ is a tunable positive real number applied over the entire study region. A large $\lambda$ makes the correction uniform within a low-resolution pixel, whereas a small $\lambda$ makes it diverse ($\lambda \rightarrow 0$ is equivalent to the original algorithm).

We compared the results from (1) the original algorithm, (2) the inverse-direction super-resolution algorithm without distance regularization (i.e., the first modification), and (3) the inverse-direction super-resolution algorithm with distance regularization (i.e., the first and the second modification). For simplicity, hereafter we call them Algorithm 1, Algorithm 2, and Algorithm 3, respectively. This comparison will clarify how our algorithm refinement improves the super-resolution results.

To summarize, the benefit of the sensor-driven algorithm over other existing approaches is explicit consideration of the PSF, and radiometric correction weighted by the Mahalanobis distance. The sensor-driven algorithm (with our improvement) may be useful for thermal super-resolution research in the context of physical consistency.

## Study site and data processing

The study site is centered around Tsukuba City, Ibaraki, Japan (36.049N-36.459N, 139.856E-140.353E; Fig 4). The region includes urban and suburban areas of Tsukuba and several neighboring cities; Mount Tsukuba, which is covered by a mixed needleleaf and broadleaf forest; and a part of Lake Kasumigaura, the second-largest inland waterbody in Japan. Rice paddy fields and croplands are distributed along several narrow river channels. According to the land cover map provided by the Japan Aerospace Exploration Agency (JAXA) [34], there are also a few grassland areas. The spatial resolution of the land cover map is 250 m. The overall accuracy and kappa coefficient have been reported as 78.0% and 0.745, respectively [34].

We searched for a cloud-free scene acquired by MODIS and ASTER simultaneously, and the scene on 24 September 2001 was selected for use. Level 1B calibrated radiances (MOD02QKM for bands 1 and 2, MOD02HKM for bands 3–7, and MOD021KM for bands 31 and 32) were downloaded via the Level-1 and Atmosphere Archive and Distribution System from the Land Processes Distributed Active Archive Center website [35]. To treat images with equally spaced meter scales, all images were projected onto the UTM 54 projection with a WGS84 ellipsoid by nearest neighbor resampling. For simplicity, super-resolution processing was conducted with images in the radiance scale (W/m$^2$/str/μm), including thermal bands. If necessary, thermal radiance can be translated into brightness temperature $T_{\mathrm{b}}$ (K) by Planck's

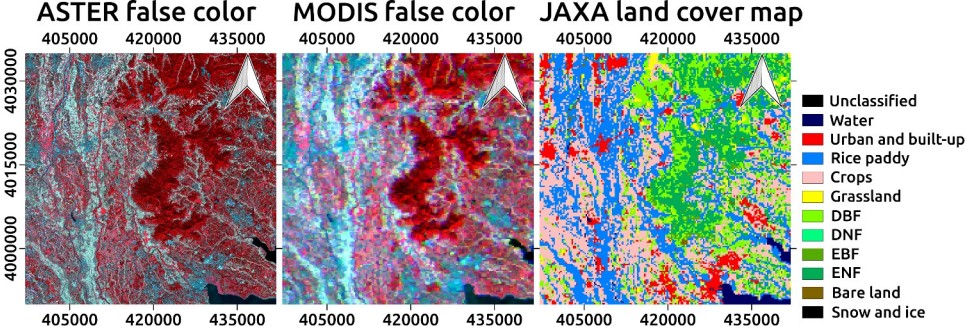

**Fig 4. Reference satellite data and land cover map for the study site.** (Left) False color image taken by ASTER (2001/09/24), (center) that taken by MODIS, and (right) JAXA land cover map degraded to 250-m resolution. All images have a UTM 54 projection with a WGS84 ellipsoid. Land cover category abbreviations: DBF, deciduous broadleaf forest; DNF, deciduous needleleaf forest; EBF, evergreen broadleaf forest; ENF, evergreen needleleaf forest.

law:

$$T_b = \frac{hc/kl}{\ln(2hc^2/l^5 B_l + 1)}, \tag{10}$$

where $h = 6.626 \times 10^{-34}$ J s is the Planck constant, $c = 2.988 \times 10^8$ m/s is the speed of light, $k = 1.380 \times 10^{-23}$ J/K is the Boltzmann constant, $l$ is the wavelength (m), and $B_l$ is the radiance (W/m$^2$/str/m) at the wavelength.

For reference, ASTER Level-3A radiance data on the same day were downloaded via the METI AIST satellite Data Archive System website [36]. The data were also projected on the UTM 54 projection with a WGS84 ellipsoid. The band correspondence between ASTER and MODIS is summarized in Table 1.

In the research performed by Tonooka [26], image coregistration between bands was not implemented because the author assumed that the accuracy of the inter-telescope registration of the data used (ASTER) was sufficient for the algorithm. However, data-driven coregistration is desirable before integrating multiple images (e.g., [12]). Therefore, we implemented image coregistration using the phase-only correlation (POC) approach [37]. Specifically, reference images (i.e., ASTER) were coregistered via POC between bands first. Then each MODIS band was coregistered by comparing it with the corresponding ASTER band (Table 1) via POC. This ensured the elimination of uncertainty caused by inconsistent MODIS inter-band registration during the super-resolution process, and inter-sensor registration between MODIS and ASTER during validation.

The MODIS PSF to simulate spatial degradation of the higher-resolution signal can be modeled by the convolution of a triangular function along the across-track direction and a Gaussian function [38, 39]. The former represents the detector response [40], and the latter represents optical blurring [38]. The PSF was considered to be a weighting function of spatial degradation within a square low-resolution pixel, which includes $v \times v$ high-resolution pixels ($v$ is the number of pixels along a column or row). The triangular function can be expressed as follows, by considering geometric transformation of the coordinates within a low-resolution pixel:

$$PSF_{tri}(i,j) =$$

$$\begin{cases} 0, & \text{if}\,|v(i,j)| > \dfrac{v}{2}\,\text{or}\,|u(i,j)| > \dfrac{v}{2} \\[2mm] -\dfrac{2u(i,j)}{v} + 1, & \text{if}\,|v(i,j)| < \dfrac{v}{2}\,\text{and}\, 0 \le u(i,j) \le \dfrac{v}{2} \\[2mm] \dfrac{2u(i,j)}{v} + 1, & \text{if}\,|v(i,j)| < \dfrac{v}{2}\,\text{and}\, -\dfrac{v}{2} \le u(i,j) < 0, \end{cases} \tag{11}$$

**Table 1. Characteristics of MODIS bands and correspondence with reference ASTER bands.**

| MODIS band | Description | MODIS wavelength (nm) | MODIS original spatial resolution (m) | ASTER band | ASTER wavelength (nm) | ASTER original spatial resolution (m) |
|---|---|---|---|---|---|---|
| 1 | Red | 620–670 | 250 | 2 | 630–690 | 15 |
| 2 | Near infrared | 841–876 | 250 | 3 | 760–860 | 15 |
| 3 | Blue | 459–479 | 500 | 1 | 520–600 | 15 |
| 4 | Green | 545–565 | 500 | 1 | 520–600 | 15 |
| 6 | Short-wave infrared | 1628–1652 | 500 | 4 | 1600–1700 | 30 |
| 7 | Short-wave Infrared | 2105–2155 | 500 | 5 | 2145–2185 | 30 |
| 31 | Thermal | 10,780–11,280 | 1000 | 14 | 10,950–11,650 | 90 |
| 32 | Thermal | 11,770–12,270 | 1000 | 14 | 10,950–11,650 | 90 |

$$u(i,j) = \frac{ai - j}{\sqrt{a^2 + 1}}, v(i,j) = \frac{i + aj}{\sqrt{a^2 + 1}}, \qquad (12)$$

where $i$ and $j$ are the high-resolution pixel coordinates in a low-resolution pixel; $u(i,j)$ and $v(i,j)$ are those for the cross-track and along-track directions, taking the center of the image as the origin; and $a$ is the inclination of the along-track direction measured on the $i$-$j$ coordinates, which was set to 5.357 by checking geolocation information in the MODIS data (MOD03 [35]).

The Gaussian function was

$$PSF_{gau}(i,j) = \exp\left(-\frac{i^2 + j^2}{2(v\sigma)^2}\right), \qquad (13)$$

where $\sigma$ determines the standard deviation of the Gaussian function, which was set to 0.2 by referring to [38, 39].

Then, the total PSF was

$$PSF_{MODIS}(i,j) = \frac{PSF_{tri}(i,j) \times PSF_{gau}(i,j)}{\sum_{(i,j)\subseteq lowres} PSF_{tri}(i,j) \times PSF_{gau}(i,j)}. \qquad (14)$$

Examples of each PSF are shown in Fig 5.

Via the super-resolution algorithm, 250-m MODIS thermal images (bands 31 and 32) were retrieved, which were validated by the corresponding ASTER band 14. To this end, the ASTER image was degraded to 250-m resolution using the MODIS PSF. The correlation coefficient (CC) and root mean squared error (RMSE) between the MODIS and ASTER images were calculated for the three types of algorithms (the original algorithm, and the proposed algorithm with and without distance regularization) to investigate the effect of our refinement. The Relative Dimensionless Global Error (ERGAS) index and peak signal-to-noise ratio (PSNR) [41] were also checked to analyze the accuracy of spectral and spatial reconstruction, respectively. Since the quantization of the thermal reference data (ASTER) is a 12-bit process [42], the maximum value is 4095 (equivalent to a radiance of 21.39 W/m$^2$/str/m), which was used for

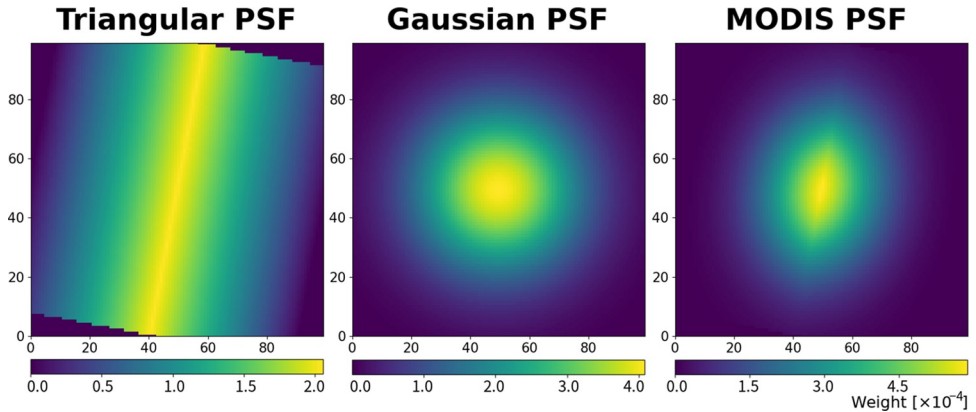

**Fig 5. Simulated Point Spread Function (PSF) for 100×100 pixels (i.e., $v$ = 100).** (left) Triangular weighting function for sensor PSF, (center) Gaussian weighting function for optical PSF, and (right) combined PSF for a Moderate Resolution Imaging Spectroradiometer (MODIS) observation.

calculating the PSNR. Spatial patterns (i.e., images) and the basic statistics for the radiance were also checked among the reference, retrieved, and original data.

## Results

The regularization parameter λ was determined as 0.002 based on tuning repeated twice, to give the first super-resolution process the best performance (i.e., the least RMSE and the best CC; Fig 6). On this basis, the validation with ASTER images for Algorithms 1, 2, and 3 is shown in Table 2. Because the relative spectral responses are different even between the corresponding bands in MODIS and ASTER images, it is natural that there is a systematic bias in radiance. Apart from the inevitable bias, the best performance is achieved with our proposed Algorithm 3, which produced the highest CC and PSNR, and the lowest RMSE and ERGAS.

More importantly, our proposed Algorithm 3 shows the best statistical consistency with the original MODIS thermal data, and as a result, also with the ASTER data (as clearly seen in Table 3). The original Algorithm 1 creates both physically impossible negative radiance and implausibly high radiance. Only the average values were acceptable because of the offset correction. Our proposed algorithm without regularization (Algorithm 2) shows better results than Algorithm 1, without any physically impossible values. However, with the appropriate regularization (Algorithm 3), the statistical consistency with the original MODIS and ASTER images increased further, not only for the average values, but also for the minimum and maximum values. Interestingly, the standard deviation of the retrieved result with Algorithm 3 is more consistent with that of the reference data (ASTER) than that of the original MODIS data.

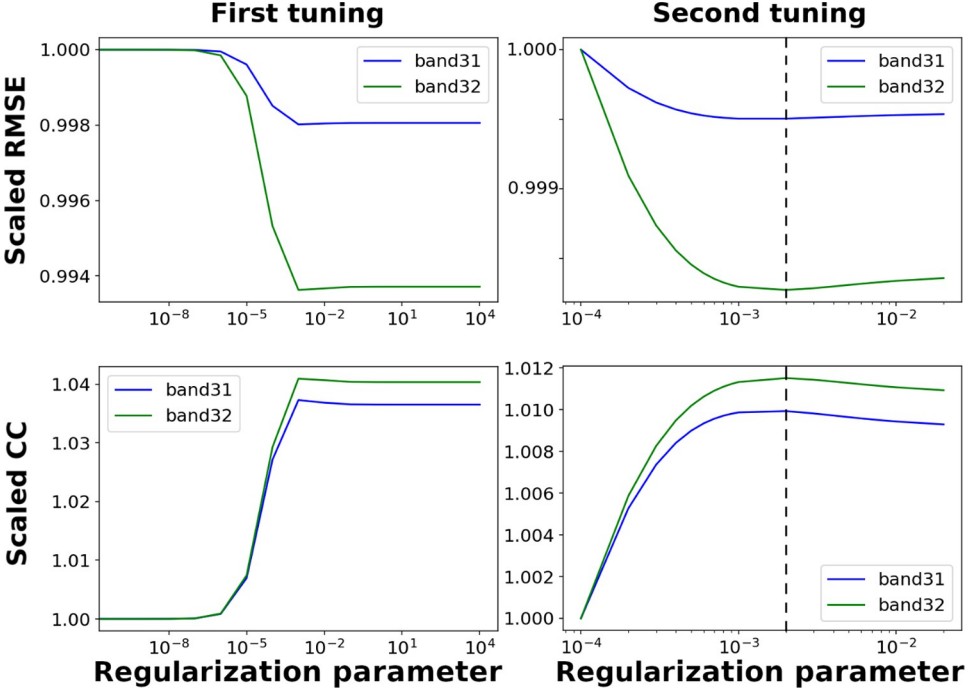

**Fig 6. Tuning of the regularization parameter λ in Algorithm 3.** (left column) Root mean squared error (RMSE) and correlation coefficient (CC) for a wide range (from 0 to $10^4$) and (right column) RMSE and CC for a narrow range (from $10^{-4}$ to $2.0 \times 10^{-2}$). Both tunings were performed with the first super-resolution image (i.e., retrieval of 500-m thermal images), and the common λ was used for the second super-resolution process. Each x-axis is log-scale, whereas each y-axis is scaled by the RMSE or CC for λ = 0. The dashed line marks λ = 0.002.

**Table 2. Correlation Coefficient (CC), Root Mean Squared Error (RMSE), and Peak Signal-to-Noise Ratio (PSNR) for each band [31, 32], and Relative Dimension-less Global Error (ERGAS) between the results from the three types of super-resolution algorithms and ASTER radiance.** CC and PSNR: larger is better; RMSE and ERGAS: smaller is better.

| Accuracy criteria | Algorithm 1 | Algorithm 2 | Algorithm 3 |
|---|---|---|---|
| CC for band 31 | 0.366 | 0.533 | 0.644 |
| RMSE / relative RMSE (%) for band 31 | 1.545 / 16.79% | 1.468 / 15.95% | 1.447 / 15.72% |
| PSNR for band 31 (dB) | 22.77 | 23.28 | 23.40 |
| CC for band 32 | 0.351 | 0.514 | 0.626 |
| RMSE / relative RMSE (%) for band 32 | 0.882 / 9.584% | 0.772 / 8.384% | 0.739 / 8.030% |
| PSNR for band 32 (dB) | 27.57 | 28.86 | 29.24 |
| ERGAS | 3.418 | 3.186 | 3.121 |

Fig 7 shows the stepwise enhancement of the spatial resolution in MODIS thermal images by Algorithm 3. Although the image contains some noisy patterns that are probably errors from our algorithm, textual details are certainly retrieved, especially around the boundaries of major land features such as Lake Kasumigaura and Mount Tsukuba. Compared with the land cover types (Fig 4), urban and built-up regions tend to show a brightness temperature higher than that of neighboring areas. Low brightness temperatures over water surfaces and forest areas are reasonable given the abundant evapotranspiration and aerodynamic features. Focusing on the forest area, the northeast part is hotter than the other area, which is probably due to the difference in altitude.

An almost similar spatial pattern can even be retrieved using Algorithms 1 and 2 (Fig 8). However, careful comparison shows that Algorithm 2 tends to generate slightly more noisy patterns and lower contrast than Algorithm 3, and that Algorithm 1 generates a few pixels having a negative brightness temperature (shown by small red points), which also confirms the results in Table 3.

## Discussion

We revisited the sensor-driven approach for thermal image super-resolution and investigated its applicability to a complex landscape with urban and suburban regions. The sensor-driven algorithm [26] with our modification refined the statistical consistency of the retrieved MODIS images (250-m resolution) with the original MODIS images and with the reference ASTER images (Tables 2 and 3). Refinement of the algorithm structure (from Algorithm 1 to 2) improved the accuracy of the super-resolution process (Table 2): in Algorithm 1, the first super-resolution process relies only on bands 1 and 2, which is likely to cause substantial

**Table 3. Basic statistics over the entire study region between the results from the three super-resolution algorithms, original MODIS image (1-km resolution), and ASTER image.**

| Radiance statistics (W/m$^2$/str/μm) | Algorithm 1 | Algorithm 2 | Algorithm 3 | Original MODIS | ASTER (band 14) |
|---|---|---|---|---|---|
| Band 31 minimum | -29.83 | 4.281 | 7.959 | 9.691 | 8.084 |
| Band 32 minimum | -27.98 | 4.685 | 7.694 | 9.100 | |
| Band 31 maximum | 27.44 | 15.04 | 11.88 | 11.23 | 11.14 |
| Band 32 maximum | 24.83 | 14.06 | 10.98 | 10.41 | |
| Band 31 average | 10.61 | 10.62 | 10.62 | 10.62 | 9.202 |
| Band 32 average | 9.876 | 9.881 | 9.881 | 9.881 | |
| Band 31 standard deviation | 0.674 | 0.450 | 0.371 | 0.290 | 0.352 |
| Band 32 standard deviation | 0.588 | 0.391 | 0.320 | 0.248 | |

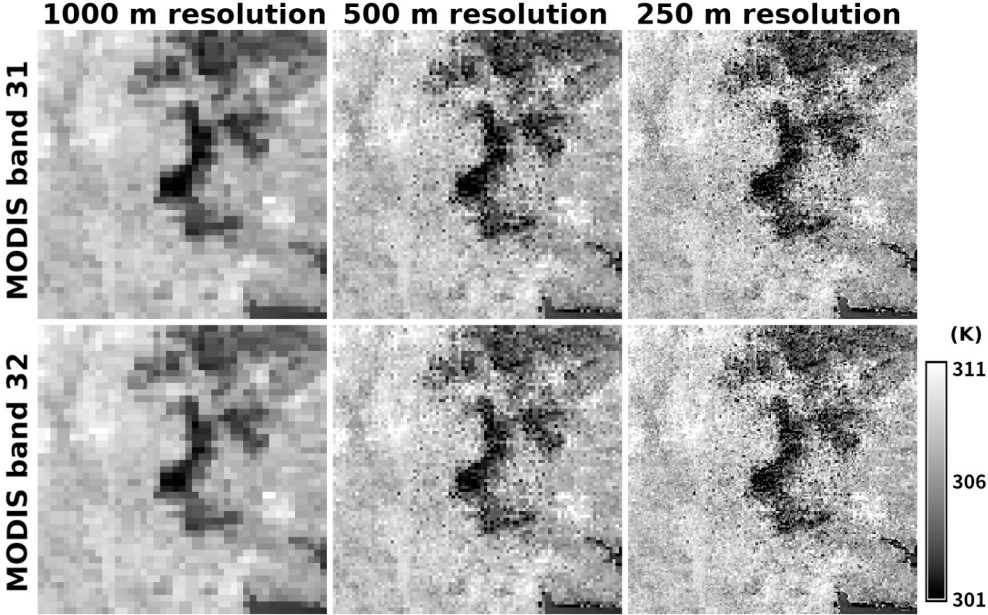

**Fig 7. Improvement in spatial resolution of MODIS thermal infrared bands by the proposed super-resolution Algorithm 3.** (left column) Original 1000-m resolution images, (center column) result from the first super-resolution process, and (right column) result from the second super-resolution process for (top row) MODIS band 31 and (bottom row) MODIS band 32. The radiance value was converted into brightness temperature by Eq 10.

uncertainty in the first super-resolution retrieval. The uncertainty probably propagates to the second super-resolution retrieval, resulting in enhanced uncertainty in the whole super-resolution process. In addition, the somewhat arbitrary hyperparameter *w* is likely to make the original algorithm too complex to obtain best-tuned results. Algorithm 2 was likely to address such issues, and was further improved by the introduction of a regularization term for the Mahalanobis distance (i.e., Algorithm 3). The standard deviation in the retrieved MODIS images using our algorithm (Algorithm 3) is more consistent with the ASTER images than with the original MODIS image with 1-km resolution. This suggests that contrasting features (i.e., spatial details) are captured by the super-resolution process, which are missed in the original MODIS image having a coarser resolution.

Retrieved thermal images well captured specific features of different land covers. In particular, urban areas (Fig 4) tend to show high brightness temperatures, suggesting an urban heat island phenomenon [43]. Statistics for each type of land cover showed that the mean thermal radiance (or brightness temperature) in urban regions is the highest among the categories (Table 4), quantitatively confirming the urban heat island phenomenon. The thermal radiance and super-resolution accuracy were similar between paddy and crop categories distributed around suburban regions. This is understandable because water in rice paddy fields should be drained in preparation for harvest in this season (September), creating similar thermal properties to crop fields before and after harvest. For further discussion of the thermal structure in an urban and suburban area, the land surface temperature [44] rather than the radiance or brightness temperature may be more suitable, although it requires additional work to estimate the thermal emissivity precisely over heterogeneous artificial materials [45, 46]. The highest accuracy (PSNR and ERGAS) was observed for the urban category, suggesting that this algorithm is suitable for obtaining super-resolution in urban landscapes. Poor accuracy was obtained in the water and forest categories. For the water category, the weak correspondence between the

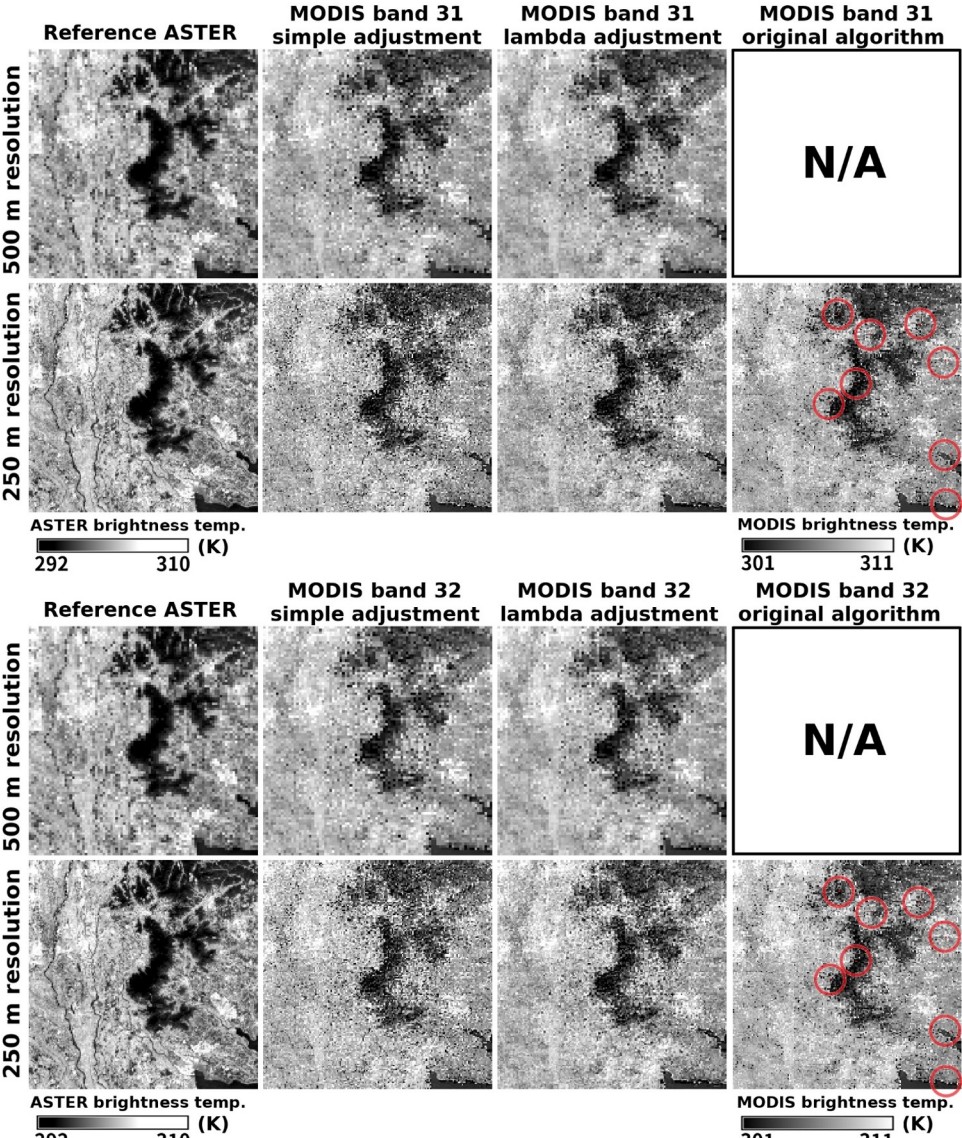

**Fig 8. Comparison of maps obtained by the three algorithms.** The upper panel containing 8 images displays MODIS band 31 results, and the lower one displays MODIS band 32 results. For each panel, from the far left column: reference ASTER images, MODIS retrieved by Algorithm 2, Algorithm 3, and Algorithm 1, respectively. The top row shows the 500-m resolution retrieval (first super-resolution images for Algorithms 2 and 3), and the bottom row shows the 250-m resolution retrieval. Algorithm 1 (original algorithm) cannot retrieve a 500-m resolution map. The red pixels (encircled by red circles for visibility) in the maps retrieved by Algorithm 1 indicate negative brightness temperatures.

thermal properties and optical spectra probably makes it difficult to conduct an accurate super-resolution process. The reason for the poor accuracy for the forest categories can be largely attributed to the temperature differences at different altitudes around Mount Tsukuba. Adding altitude data (i.e., digital elevation model) when calculating the Mahalanobis distance and clustering homogeneous pixels may improve the results for such mountainous regions.

Improvement in the accuracy indices by our algorithm refinement was statistically confirmed for all land cover categories (Table 5). The Wilcoxon signed-rank test over the categories ($n = 7$) indicated that Algorithm 3 showed a smaller ERGAS and a greater PSNR than

**Table 4. Statistics for super-resolved thermal data and accuracy indices for Algorithm 3 for each land-cover type.** The land-cover type was determined using previously reported data [34], while unclassified and snow/ice categories were excluded. Note that the four forest categories (see Fig 4) were integrated into the one category. Radiance (W/m$^2$/str/μm) with standard deviation, $T_b$ (brightness temperature; K) with standard deviation, PSNR, ERGAS, and N (number of sample pixels) are listed.

| Categories | B31 radiance / $T_b$ | B32 radiance / $T_b$ | B31 PSNR | B32 PSNR | ERGAS | N |
|---|---|---|---|---|---|---|
| All | 10.62±0.37 / 306.9±2.5 | 9.881±0.320 / 307.1±2.5 | 23.40 | 29.23 | 3.121 | 32400 |
| Water | 10.16±0.41 / 303.8±2.8 | 9.453±0.373 / 303.7±3.0 | 23.03 | 28.28 | 3.501 | 552 |
| Urban | 10.91±0.26 / 308.9±1.8 | 10.13±0.23 / 309.0±1.8 | 24.04 | 31.01 | 2.712 | 2285 |
| Paddy | 10.74±0.29 / 307.7±1.9 | 9.985±0.253 / 307.9±2.0 | 23.70 | 30.11 | 2.921 | 9027 |
| Crop | 10.73±0.26 / 307.7±1.7 | 9.978±0.227 / 307.9±1.8 | 23.55 | 29.83 | 2.990 | 9854 |
| Grassland | 10.58±0.34 / 306.6±2.3 | 9.847±0.298 / 306.9±2.3 | 23.98 | 30.13 | 2.872 | 589 |
| Forest | 10.35±0.38 / 305.1±2.6 | 9.662±0.324 / 305.4±2.6 | 22.86 | 27.88 | 3.527 | 9914 |
| Barren | 10.51±0.44 / 306.2±3.1 | 9.803±0.381 / 306.5±3.0 | 23.58 | 28.84 | 3.105 | 179 |

Algorithm 2 with statistical significance ($p < 0.01$), and Algorithm 2 showed a smaller ERGAS and greater PSNR than Algorithm 1 with statistical significance ($p < 0.01$). Therefore, for all categories, Algorithm 2 was better than Algorithm 1, and Algorithm 3 was better than both Algorithms 1 and 2.

Artificial coloring and metal composition of the surface would also influence the estimation of radiance or brightness temperature using our super-resolution algorithm, especially when creating the typical spectral pattern, which relies on the spectral link between the optical and thermal domains. In practice, the impact of the uncertainty in the typical spectral pattern on the super-resolution is limited because most of the thermal radiance $g_{\text{high,k'}}$ is retrieved from neighboring homogeneous pixels, rather than the typical spectral pattern extracted by clustering (Fig 9G and 9H). Therefore, as long as a spectrally similar target with 500-m spatial homogeneity in the second super-resolution image can be obtained around the pixel of interest, the resulting image is likely to be reliable. This condition may sometimes be too strict for heterogeneous landscapes including urban and suburban areas, and thus retrieval has uncertainty in such regions. In fact, the Mahalanobis distance for the retrieval is small (i.e., high reliability in the retrieval) over the relatively homogeneous forest region and water body apart from the lake shore (Fig 9D and 9F), but not in the urban and suburban areas, and the boundary the land covers (e.g., the shore of Kasumigaura Lake) with less homogeneity (Fig 9A–9C). However, even in such cases, the offset correction with regularization at least ensures statistical consistency in the final retrieved value $g'_{\text{high,k'}}$. Super-resolution with higher-resolution data (e.g., ASTER) may further mitigate uncertainty arising from such spatial heterogeneity.

There is still room to improve our algorithm regarding the visualizability of the retrieved image. Textural details, such as narrow river channels and mixed landscapes of crop, forest, urban, and suburban areas (Fig 4) are hidden behind the noisy patterns generated by the

**Table 5. Comparison of accuracy indices (ERGAS: Smaller is better; PSNR: Greater is better) for different algorithms and land-cover types.**

| Accuracy indices | ERGAS | | | PSNR31 | | | PSNR32 | | |
|---|---|---|---|---|---|---|---|---|---|
| Algorithm | 1 | 2 | 3 | 1 | 2 | 3 | 1 | 2 | 3 |
| Water | 3.831 | 3.571 | 3.501 | 22.51 | 22.93 | 23.03 | 26.71 | 27.88 | 28.28 |
| Urban | 2.930 | 2.788 | 2.712 | 23.56 | 23.87 | 24.04 | 29.47 | 30.44 | 31.01 |
| Paddy | 3.137 | 2.986 | 2.921 | 23.27 | 23.57 | 23.70 | 28.74 | 29.66 | 30.11 |
| Crop | 3.094 | 3.040 | 2.990 | 23.34 | 23.45 | 23.55 | 29.18 | 29.52 | 29.83 |
| Grassland | 3.195 | 3.042 | 2.872 | 23.29 | 23.62 | 23.98 | 28.34 | 29.10 | 30.13 |
| Forest | 4.055 | 3.599 | 3.527 | 21.94 | 22.73 | 22.86 | 25.86 | 27.56 | 27.88 |
| Barren | 4.632 | 3.186 | 3.105 | 20.80 | 23.49 | 23.58 | 23.61 | 28.19 | 28.84 |

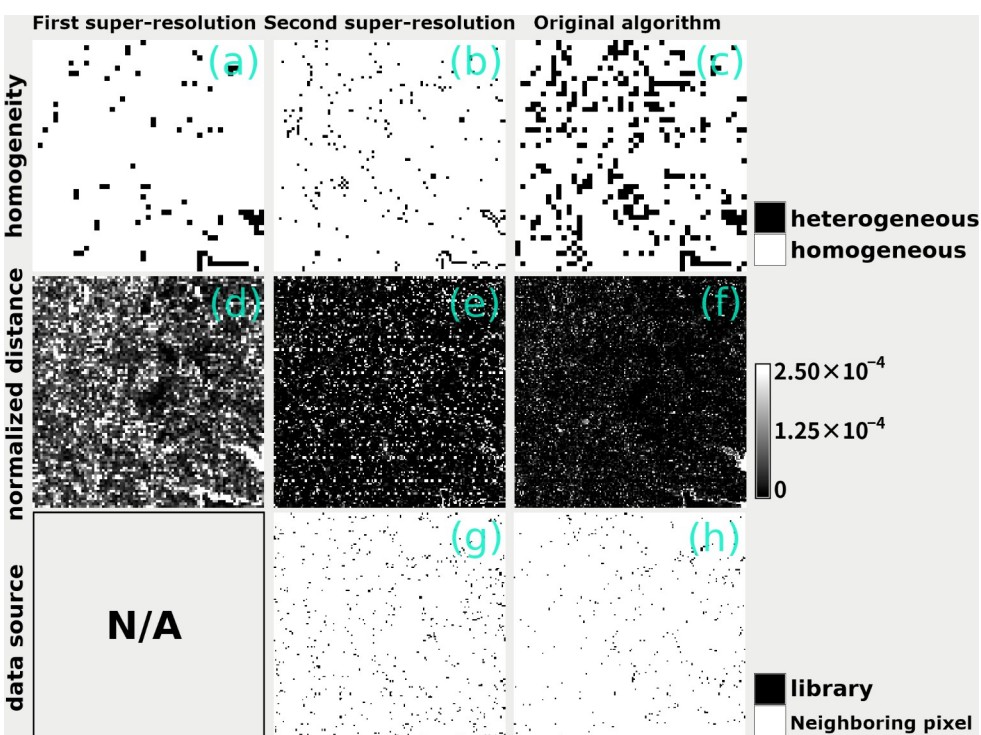

**Fig 9. Maps describing characteristics of super-resolution retrieval.** From the top row, pixel homogeneity, Mahalanobis distance, and data sources (the typical spectral pattern or neighboring pixel used in the retrieval process) are displayed. The left and center columns show the maps for the first and the second super-resolution retrievals by our proposed Algorithm 3, respectively. The right column shows retrieval by the original Algorithm 1.

algorithm (Fig 7). Due to the inherent feature of the twofold super-resolution retrieval, the noise generated in the first super-resolution image should inevitably affect the second super-resolution image. In fact, a distributed spatial pattern of the large Mahalanobis distance (i.e., less reliable retrieval) is observed in the distance map in the second super-resolution image (Fig 9E), which can be attributed to the noise generated by the first super-resolution retrieval.

Data-driven approaches, including traditional pan-sharpening [18, 19], kernel-driven methods [20–22], and machine learning [23–25], may have an advantage from the viewpoint of visualizability. Therefore, comparison of the sensor-driven approach with such data-driven approaches and/or their integrated use will be important future work. Especially, the offset correction, which is an important part of our algorithm, may be added to other data-driven approaches to improve the statistical consistency, while keeping each algorithm straightforward.

## Conclusion

To improve the spatial resolution of thermal satellite images, we revisited a sensor-driven super-resolution algorithm and investigated its applicability to a complex landscape with urban and suburban regions. The algorithm explicitly considers the sensor blurring effect using a point spread function, and ensures radiometric consistency with the original thermal image during high-resolution thermal image retrieval, both of which are not generally taken into consideration in existing approaches such as machine learning and kernel-driven methods. We also introduced modification to the original sensor-driven algorithm to enhance the statistical consistency of the super-resolution results, including making the algorithm structure

more straightforward, and introducing regularization term when calculating the Mahalanobis distance.

The original sensor-driven algorithm (Algorithm 1) and two refined versions (Algorithms 2 and 3) were applied to a cloud-free MODIS scene to enhance the thermal (1 km) resolution to the optical (250 m) resolution, and were validated against the corresponding high-resolution thermal image (ASTER). The validation result showed that the refined sensor-driven algorithm can downscale the MODIS image to 250-m resolution, while maintaining a high statistical consistency with the original MODIS and ASTER images. Part of our algorithm, such as radiometric offset correction based on the Mahalanobis distance, may be integrated with other existing approaches in the future.

## Acknowledgments

This research was supported by Research Laboratory on Environmentally-conscious Developments and Technologies (E-code) of the National Institute of Advanced Industrial Science and Technology. Publicly available datasets were analyzed in this study. The data can be found on the JAXA land cover map [34], MODIS [35], and ASTER [36] websites.

## Author Contributions

**Conceptualization:** Hiroki Mizuochi, Koki Iwao, Satoru Yamamoto.

**Data curation:** Hiroki Mizuochi.

**Formal analysis:** Hiroki Mizuochi.

**Investigation:** Hiroki Mizuochi.

**Methodology:** Hiroki Mizuochi.

**Project administration:** Koki Iwao.

**Software:** Hiroki Mizuochi.

**Validation:** Hiroki Mizuochi.

**Visualization:** Hiroki Mizuochi.

**Writing – original draft:** Hiroki Mizuochi.

**Writing – review & editing:** Koki Iwao, Satoru Yamamoto.

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
