## [Decision Letter · Decision Letter 0]

26 Jan 2022

PONE-D-21-32571

Thermal remote sensing over heterogeneous urban and suburban landscapes using sensor-driven super-resolution

PLOS ONE

Dear Hiroki,

Thank you for submitting your manuscript titled, ‘Thermal remote sensing over heterogeneous urban and suburban landscapes using sensor-driven super-resolution (Ref. No.: PONE-D-21-32571) to PLOS ONE. The manuscript has been reviewed. After careful consideration, we find that the manuscript has some merit with some interesting results, but it needs a major revision to address reviewers’ comments and fully meet PLOS ONE publication criteria. You can find reviewers’ comments at the bottom of this letter.

We invite you to submit a revised version of the manuscript. The changes required in the manuscript are very significant and require you to respond fully. We will send your revised manuscript for further external review. Therefore, we strongly recommend addressing concerns raised in full.

Be sure to address:

1. Restructure your introduction, particularly organizing the problem statement 

2. Quality and extent of results.

3. Revise the Discussion section for clarity.

We look forward to receiving your revised manuscript.

Kind regards,

Dr. Kunwar K. Singh

Academic Editor

PLOS ONE

Journal Requirements:

Reviewers' comments:

Reviewer's Responses to Questions

**Comments to the Author**

1. Is the manuscript technically sound, and do the data support the conclusions?

Reviewer #1: Partly

Reviewer #2: Yes

2. Has the statistical analysis been performed appropriately and rigorously? 

Reviewer #1: No

Reviewer #2: I Don't Know

3. Have the authors made all data underlying the findings in their manuscript fully available?

Reviewer #1: Yes

Reviewer #2: Yes

4. Is the manuscript presented in an intelligible fashion and written in standard English?

Reviewer #1: Yes

Reviewer #2: Yes

5. Review Comments to the Author

Reviewer #1: Dear authors:

I have found your research article quite interesting. I have carefully read your proposal. I have some questions and explanations I would like you to clarify for me.

Your research tries to recover and deepen a method to improve the resolution of thermal images from a sensor point of view (sensor-driven), as opposed to other approaches (data-driven).

The preceding methodology and your proposals for improvement are well defined. Except in the figures shown:

The order of the defined steps is not clear. In figure 1, there is a red arrow linking both steps of superresolution. Please clarify its meaning. In figure 2, MODIS B1,2(1Km) and (500 m) refer to one super-resolution step when they are images with different resolutions. The reader may be confused about the use of both images with different resolutions at the same super-resolution step.

In line 103, (step 3) it is not clear what the meaning of "instant spectral library" is. I would thank the authors for clarifying this concept.

In line 145, the authors refer to a super-resolution step using different bands simultaneously. The authors do not make clear how these many bands are handled. Clarification about how the bands fuse with the thermal image will be useful.

My major concern in this work, in the Results section, is about the statistics the authors use to determine the quality of the super-resolution transformation. RMSE is not a good index of image quality. It has been proved that very different images can have the same RMSE value. The same is true for the correlation coefficient (CC). There are other more appropriate image quality indices such as the ERGAS or the spectral mapping angle (SAM) for spectral reconstruction. If you want to analyze the spatial reconstruction, Peak Signal to Noise Ratio (PSNR) will measure the quality of this process. Please refer to those indices or justify the use of RMSE and CC properly.

I will be grateful if the authors could clarify these comments. I encourage them because I consider their work an important contribution to thermal image resolution enhancement literature.

Reviewer #2: The article presents a review of a methodology to obtain surface temperature values based on spatially resampled MODIS sensor data. In its presentation, the main objective of the article is to demonstrate the application of the method for urban and suburban areas, however the results and discussions presented do not respond to this expectation, which is the main point of the improvements that the article needs.

Below are some excerpts that indicate adjustments and additions to improve the understanding of the article.

In line 44, I suggest improving the presentation of the characteristics of the thermal images, perhaps presenting a table or in the text, detailing the resolutions of each sensor

In line 75, the authors seek to show the potential application of the algorithm, however the work does not show other results obtained with the use of the first algorithm that justify changes in it. I suggest that they present other works that analyzed the application of the 1 algorithm or made comparisons with other sensors

In line 87, In the final paragraph of the Introduction, I suggest adding the objectives and results expected by the work.

In line 103, in which he presents the routine for image treatment, I suggest improving the flowchart presented, using a number with stage indications, as this way the reader understands which steps the objective of the work changes and facilitates the replication of the work

In line 135, I suggest rescuing the reason for the proposed algorithm modification

In line 144, Here again rescue the routine flowchart of the algorithm indicating where changes occurred and to present a new proposal

In line 327, I recommend that the authors seek to contextualize the discussion of the results obtained by showing the temperature results according to the land uses present in the analyzed section. It is important and necessary to confirm statistically whether there were differences or changes in temperatures by observing the behavior of surface temperature in vegetated environments and urban spaces

6. PLOS authors have the option to publish the peer review history of their article (what does this mean?). If published, this will include your full peer review and any attached files.

Reviewer #1: No

Reviewer #2: No

---

## [Author Response · Author response to Decision Letter 0]

16 Feb 2022

Reviewer #1: Dear authors:

I have found your research article quite interesting. I have carefully read your proposal. I have some questions and explanations I would like you to clarify for me. Your research tries to recover and deepen a method to improve the resolution of thermal images from a sensor point of view (sensor-driven), as opposed to other approaches (data-driven). The preceding methodology and your proposals for improvement are well defined. Except in the figures shown:

The order of the defined steps is not clear. In figure 1, there is a red arrow linking both steps of superresolution. Please clarify its meaning. In figure 2, MODIS B1,2(1Km) and (500 m) refer to one super-resolution step when they are images with different resolutions. The reader may be confused about the use of both images with different resolutions at the same super-resolution step.

Thank you for your encouraging and constructive comments. Based on this and the comment from reviewer 2, we have added a new figure (Fig. 1) and explanation (Lines 124-130) to clarify the flowchart and corresponding steps. We also added a step-by-step explanation in Fig. 2 (Lines 131-149) and Fig. 3 (Line 178). For each super-resolution process, high-resolution images (fhigh,k), degraded high-resolution images (flow,k), and low-resolution images (glow,k’) are needed, and then super-resolution images (ghigh,k’) are output. For example, MODIS B1,2 (1 km) is flow,k, and B1,2 (500 m) is fhigh,k in Fig. 3 (original Fig. 2).

In line 103, (step 3) it is not clear what the meaning of "instant spectral library" is. I would thank the authors for clarifying this concept.

We decided not to use this ambiguous term, but explicitly explain the concept (Line 112) as follows: “Step 3) Make a typical spectral pattern (i.e., correspondence between flow,k and glow,k’) by clustering the homogeneous pixels within the entire study region.”

In line 145, the authors refer to a super-resolution step using different bands simultaneously. The authors do not make clear how these many bands are handled. Clarification about how the bands fuse with the thermal image will be useful.

Please see the abovementioned comments regarding the original Figs. 1 and 2, and the new Fig. 1. The super-resolution algorithm searches for the correspondence between flow,k and glow,k’ from a neighboring pixel or the typical spectral pattern, and simultaneously positions ghigh,k’ for all bands k’.

My major concern in this work, in the Results section, is about the statistics the authors use to determine the quality of the super-resolution transformation. RMSE is not a good index of image quality. It has been proved that very different images can have the same RMSE value. The same is true for the correlation coefficient (CC). There are other more appropriate image quality indices such as the ERGAS or the spectral mapping angle (SAM) for spectral reconstruction. If you want to analyze the spatial reconstruction, Peak Signal to Noise Ratio (PSNR) will measure the quality of this process. Please refer to those indices or justify the use of RMSE and CC properly.

Based on this suggestion, we have added ERGAS and PSNR for accuracy assessment (Line 304; Table 2) and added new references [41, 42]. These indices also indicate a performance improvement with our algorithm, as was the case for the RMSE and CC. We did not use the SAM because we used the same ASTER reference data (band 14) for both MODIS bands (bands 31 and 32), and calculation of spectral similarity is not applicable in this case.

Reviewer #2: The article presents a review of a methodology to obtain surface temperature values based on spatially resampled MODIS sensor data. In its presentation, the main objective of the article is to demonstrate the application of the method for urban and suburban areas, however the results and discussions presented do not respond to this expectation, which is the main point of the improvements that the article needs.

Thank you for reviewing our manuscript. To demonstrate the results from the viewpoint of urban and suburban landscapes, we have added a new table to show the statistics related to the results (super-resolution thermal values and accuracy indices) for each land-use and land-cover type, including urban and six other categories (Table 4). This analysis, for example, quantitatively confirmed the heat island phenomenon in the urban region (Line 378), and revealed that the best super-resolution performance was observed in the urban region (Line 385). We have also added further findings of this analysis in the Discussion section (Line 378-397).

In line 44, I suggest improving the presentation of the characteristics of the thermal images, perhaps presenting a table or in the text, detailing the resolutions of each sensor

We have added a description of the degradation of the spatial resolution of thermal data with examples from MODIS and ASTER, in the Introduction (Line 36-40).

In line 75, the authors seek to show the potential application of the algorithm, however the work does not show other results obtained with the use of the first algorithm that justify changes in it. I suggest that they present other works that analyzed the application of the 1 algorithm or made comparisons with other sensors

Indeed, the applications on Mars [4] using THEMIS [27] (Line 69) and in urban and suburban areas [26] using ASTER (Line 77) were only the existing reports that we knew of, and neither provided sufficient quantitative accuracy assessment by comparison with independent validation data (Line 77). Therefore, this paper represents the first quantitative report of the performance of algorithm 1. We consider that this is a point of novelty of our research, and added an appropriate statement in the Introduction (Line 92).

In line 87, In the final paragraph of the Introduction, I suggest adding the objectives and results expected by the work.

We had already stated the research objective in the first sentence of the final paragraph (Line 80): “this work aims to investigate the potential applicability of the sensor-driven approach over a heterogeneous landscape, and to improve its primitive algorithm”. The expected result has been newly added (Line 92) based on this comment, relating to the abovementioned response to the original line 75.

In line 103, in which he presents the routine for image treatment, I suggest improving the flowchart presented, using a number with stage indications, as this way the reader understands which steps the objective of the work changes and facilitates the replication of the work

Based on this and the comment from reviewer 1, we have added a new figure to clarify the super-resolution process (Fig. 1), indicating the improved points (star symbols). A brief explanation of the overall process has also been added (Lines 124-130).

In line 135, I suggest rescuing the reason for the proposed algorithm modification

We have briefly stated the purpose of the refinement (Line 164): “To make the algorithm more straightforward and to create better radiometrically corrected results,”

In line 144, Here again rescue the routine flowchart of the algorithm indicating where changes occurred and to present a new proposal

We have added a description of how the refinements corresponded to each step (Line 166) by revisiting the flowchart as follows: “For each super-resolution process, refinement (1) concerns input-output correspondence and degraded image input, whereas refinement (2) concerns post-processing (both are indicated by a star symbol in the flowchart in Fig. 1).”

In line 327, I recommend that the authors seek to contextualize the discussion of the results obtained by showing the temperature results according to the land uses present in the analyzed section. It is important and necessary to confirm statistically whether there were differences or changes in temperatures by observing the behavior of surface temperature in vegetated environments and urban spaces

Please see the response to the general comment. We have added a statistical analysis regarding each land cover category in addition to a discussion (Lines 378-397).

---

## [Decision Letter · Decision Letter 1]

4 Mar 2022

PONE-D-21-32571R1Thermal remote sensing over heterogeneous urban and suburban landscapes using sensor-driven super-resolutionPLOS ONE

Dear Hiroki,

Thank you for submitting a revised manuscript titled, ‘Thermal remote sensing over heterogeneous urban and suburban landscapes using sensor-driven super-resolution’ to PLOS ONE. Reviewers’ have provided their feedback. The quality of manuscript has improved substantially, but it will require a minor revision to address reviewers’ comments and fully meet PLOS ONE publication criteria. You can find reviewers’ comments at the bottom of this letter.

We invite you to submit a revised version of the manuscript.

 Be sure to address:

1. Improve the statistical exploration of the results, including apply non-parametric Wilcox text

2. Emphasize gains from the modification of the algorithm.  

We would appreciate receiving your revised manuscript by Apr 18 2022 11:59PM. When you are ready to submit your revision, log on to https://pone.editorialmanager.com/ and select the 'Submissions Needing Revision' folder to locate your manuscript file.

Please include the following items when submitting your revised manuscript:A rebuttal letter that responds to each point raised by the academic editor and reviewer(s). You should upload this letter as a separate file labeled 'Response to Reviewers'.A marked-up copy of your manuscript that highlights changes made to the original version. You should upload this as a separate file labeled 'Revised Manuscript with Track Changes'.An unmarked version of your revised paper without tracked changes. You should upload this as a separate file labeled 'Manuscript'.If applicable, we recommend that you deposit your laboratory protocols in protocols.io to enhance the reproducibility of your results. Protocols.io assigns your protocol its own identifier (DOI) so that it can be cited independently in the future. For instructions see: https://journals.plos.org/plosone/s/submission-guidelines#loc-laboratory-protocols. Additionally, PLOS ONE offers an option for publishing peer-reviewed Lab Protocol articles, which describe protocols hosted on protocols.io. Read more information on sharing protocols at https://plos.org/protocols?utm_medium=editorial-email&utm_source=authorletters&utm_campaign=protocols.

We look forward to receiving your revised manuscript.

Kind regards,

Kunwar K. Singh

Academic Editor

PLOS ONE

Journal Requirements:

Reviewers' comments:

Reviewer's Responses to Questions

**Comments to the Author**

1. If the authors have adequately addressed your comments raised in a previous round of review and you feel that this manuscript is now acceptable for publication, you may indicate that here to bypass the “Comments to the Author” section, enter your conflict of interest statement in the “Confidential to Editor” section, and submit your "Accept" recommendation.

Reviewer #1: All comments have been addressed

Reviewer #2: All comments have been addressed

2. Is the manuscript technically sound, and do the data support the conclusions?

Reviewer #1: Yes

Reviewer #2: Yes

3. Has the statistical analysis been performed appropriately and rigorously? 

Reviewer #1: Yes

Reviewer #2: No

4. Have the authors made all data underlying the findings in their manuscript fully available?

Reviewer #1: Yes

Reviewer #2: Yes

5. Is the manuscript presented in an intelligible fashion and written in standard English?

Reviewer #1: Yes

Reviewer #2: Yes

6. Review Comments to the Author

Reviewer #1: I would like to thank the authors for their work in clarify my comments and suggestions about their original work. Now, in the new version, I have found this research paper clearer and the quality of the results are well established with the new statistical indices.

I am glad to propose your work for publishing.

Reviewer #2: The authors met the requests made, presenting more details about how the workflow occurs for the first algorithm and presenting clarifications regarding the modifications and improvements proposed by the work that occurred in it.

It is still necessary to improve the statistical exploration of the results obtained, demonstrating the real improvement that the modification of the algorithm brings.

I suggest application of the non-parametric Wilcox test was applied, with 95% confidence, to determine if there was a significant difference between the radiance and brightness temperature occurring in different land covers and land uses for Algorithm1, 2 and 3.

This further analysis will enrich the results presented and effectively show the gains with the modification of the algorithm

7. PLOS authors have the option to publish the peer review history of their article (what does this mean?). If published, this will include your full peer review and any attached files.

Reviewer #1: **Yes: **Javier Raimundo

Reviewer #2: No

---

## [Author Response · Author response to Decision Letter 1]

16 Mar 2022

Reviewer #1: I would like to thank the authors for their work in clarify my comments and suggestions about their original work. Now, in the new version, I have found this research paper clearer and the quality of the results are well established with the new statistical indices.

I am glad to propose your work for publishing.

We are glad to hear your very positive comment, and thank you again for reviewing our paper.

Reviewer #2: The authors met the requests made, presenting more details about how the workflow occurs for the first algorithm and presenting clarifications regarding the modifications and improvements proposed by the work that occurred in it.

It is still necessary to improve the statistical exploration of the results obtained, demonstrating the real improvement that the modification of the algorithm brings.

I suggest application of the non-parametric Wilcox test was applied, with 95% confidence, to determine if there was a significant difference between the radiance and brightness temperature occurring in different land covers and land uses for Algorithm1, 2 and 3.

This further analysis will enrich the results presented and effectively show the gains with the modification of the algorithm

Thank you for your constructive comments. We understand that this suggestion refers to the statistical significance regarding improvement of the accuracy indices (PSNR and EAGAS) among the three algorithms, and we agree that the recommended analysis would be helpful for clarifying the gains due to modification of the algorithm. Therefore, we compared the accuracy indices for each category for the different algorithms (Table 5), and performed a Wilcoxon signed-rank test using classified samples (i.e., n = 7) as suggested. The results show that Algorithm 3 had a smaller ERGAS and a greater PSNR than Algorithm 2 with statistical significance (p < 0.01), and that Algorithm 2 had a smaller ERGAS and a greater PSNR than Algorithm 1 with statistical significance (p < 0.01). Therefore, for all categories, Algorithm 2 was better than Algorithm 1, and Algorithm 3 was better than both Algorithms 1 and 2 (Line 392). We added this description to the Discussion section.

---

## [Editor Report · Decision Letter 2]

23 Mar 2022

Thermal remote sensing over heterogeneous urban and suburban landscapes using sensor-driven super-resolution

PONE-D-21-32571R2

Dear Hiroki,

We’re pleased to inform you that your manuscript has been judged scientifically suitable for publication and will be formally accepted for publication once it meets all outstanding technical requirements.

Kind regards,

Kunwar K. Singh

Academic Editor

PLOS ONE
---

## [Editor Report · Acceptance letter]

28 Mar 2022

PONE-D-21-32571R2 

Thermal remote sensing over heterogeneous urban and suburban landscapes using sensor-driven super-resolution 

Dear Dr. Mizuochi:

I'm pleased to inform you that your manuscript has been deemed suitable for publication in PLOS ONE. Congratulations! Your manuscript is now with our production department. 

Kind regards, 

on behalf of

Dr. Kunwar K. Singh 

Academic Editor

PLOS ONE